# Surviving uncertainty: A dual-path model of personal initiative affecting graduate employability

Ting Wu[1], Qin Lai[1], Nan Ma[2,3]*, Yixuan Shao[4]

**1** School of Business, Zhejiang University City College, Hangzhou, China, **2** School of Information and Electrical Engineering, Zhejiang University City College, Hangzhou, China, **3** College of Education, Zhejiang University, Hangzhou, China, **4** Department of Psychology and Behavioural Science, Zhejiang University, Hangzhou, China

* manan0831@163.com

**Data Availability Statement:** All relevant data are within the paper and its Supporting Information files.

**Funding:** This work was supported by Zhejiang Provincial Education Department Project under

## Abstract

The increasing uncertainty of our world raises important questions for university students on how they should respond to the employment challenges caused by changing environments. One of the central topics is the development of graduate employability. However, most previous research on graduate employability was undertaken in a stable environment, limiting our understanding of how graduate employability develops in a dynamic context. We have advanced the literature by introducing action theory to investigate the process of personal initiative affecting graduate employability in a period of environmental uncertainty. Using a time-lagged research design, we collected data from a sample of 229 Chinese university students and tested the hypothesized relationships. We find that personal initiative positively affects graduate employability through human and psychological capital. We further show that environmental uncertainty plays a contingent role in the above processes. Specifically, when a high level of environmental uncertainty is perceived, the positive indirect effect of personal initiative on graduate employability through either human capital or psychological capital is more likely to be strengthened. The theoretical and practical implications of these findings are also discussed.

## Introduction

When considering global issues such as the pandemic caused by the coronavirus disease of 2019 (COVID-19), the Russia-Ukraine war, and trade conflicts between major powers, the uncertainty of our world becomes even more apparent [1,2]. This uncertainty raises critical questions on how individuals, organizations, and society can prepare for its impact. For university students, such questions relate to how they may be expected to prepare to find a means of living in an environment of deteriorating economic conditions with job markets that have limited growth or are even shrinking [3,4]. Therefore, it is not surprising that the development of graduate employability has become one of the central missions of higher education institutions and government policy-makers. As shown by Abelha et al.'s (2020) systematic review, the

Grant No. Y202044350, Zhejiang Provincial Education Science Planning Project under Grant No. 2022SCG217 and 2021SB097, National Natural Science Foundation of China under Grant No. 72072058, Humanities and Social Sciences Program of the Ministry of Education under Grant No. 20YJC630235, and Research Center of Digital Transformation and Social Responsibility Management, ZUCC.

**Competing interests:** The authors have declared that no competing interests exist.

body of research that has been conducted to investigate graduate employability has steadily been growing, as the number of published studies on this topic has increased fourfold from 2010 to 2019 [5].

However, current studies investigating the development of graduate employability are mainly decontextualized; they were primarily conducted in stable settings without considering the possible constraints that high environmental uncertainty (e.g., the COVID-19 pandemic) may impose on the development of graduate employability [6,7]. As Donald et al. (2021) indicated, graduate employability is socially constructed in and from a particular context [7]. In this regard, the increased uncertainty of our world due to the COVID-19 pandemic, the Russia-Ukraine war, and trade conflicts between major powers may create contingencies, constraints, and causations that prompt researchers to introduce the context of environmental uncertainty in graduate employability studies.

In addition, most of the previous studies explored the antecedents of graduate employability from the perspective of higher education institutions or government policy-makers, ignoring the crucial impact of initiatives taken by university students themselves [8,9]. For example, Ali and Marwan (2019) suggested that higher education institutions play an essential role in understanding and developing graduates' competencies to meet the demands of professional industries. They proposed that universities could introduce work-based learning (WBL), one of the platforms providing the opportunity to gain real work experience, to help students foster graduate employability and start their careers. As another example, Wondwosen (2022) argued that government policy-makers have the primary responsibility for fostering graduate employability, as only the participation and coordinated efforts of the government at federal and regional levels can address the core issues that lead to low graduate employability, namely, questionable teacher quality, poor-quality graduates, and weak linkages with industries. Although these studies have constructed strategies to promote graduate employability at the macro-level [10,11], they have not revealed the psychological mechanisms underlying the development of graduate employability.

Therefore, a question that deserves further investigation is what determines the micro-foundations for developing graduate employability in the particular context of uncertainty. Although previous research has confirmed the direct impact or influence process of some personal factors, such as academic achievements [12], soft skills [13], and career identity [14] on graduate employability, these factors are not sufficient to help university students overcome the obstacles caused by increased environmental uncertainty. The construct of personal initiative (PI), which initially came from the field of education [15] and was later refined and developed in management [e.g., 16,17], psychology [e.g., 18–20], and entrepreneurship [e.g., 21–23], is strongly associated with positive outcomes for individuals, teams, and organizations. In particular, because entrepreneurship is characterized by high uncertainty [24–26], research has shown that teaching PI would be a better approach for boosting small businesses in West Africa than traditional methods of training [27]. Therefore, the reintroduction of PI in the field of education will contribute to understanding how graduate employability is developed in a context of uncertainty.

Taken together, the research gaps regarding graduate employability are twofold. On the one hand, most studies neglected the context of uncertainty in developing graduate employability. On the other hand, few studies have investigated the micro-foundations of graduate employability from the perspective of university students. These two research gaps eventually motivated us to raise the research question—how does university students' PI affect their graduate employability in the context of uncertainty? To address the research question, we proposed a dual-path model of PI affecting graduate employability. Specifically, we introduced an action theory to investigate how PI affects graduate employability through two types of capital

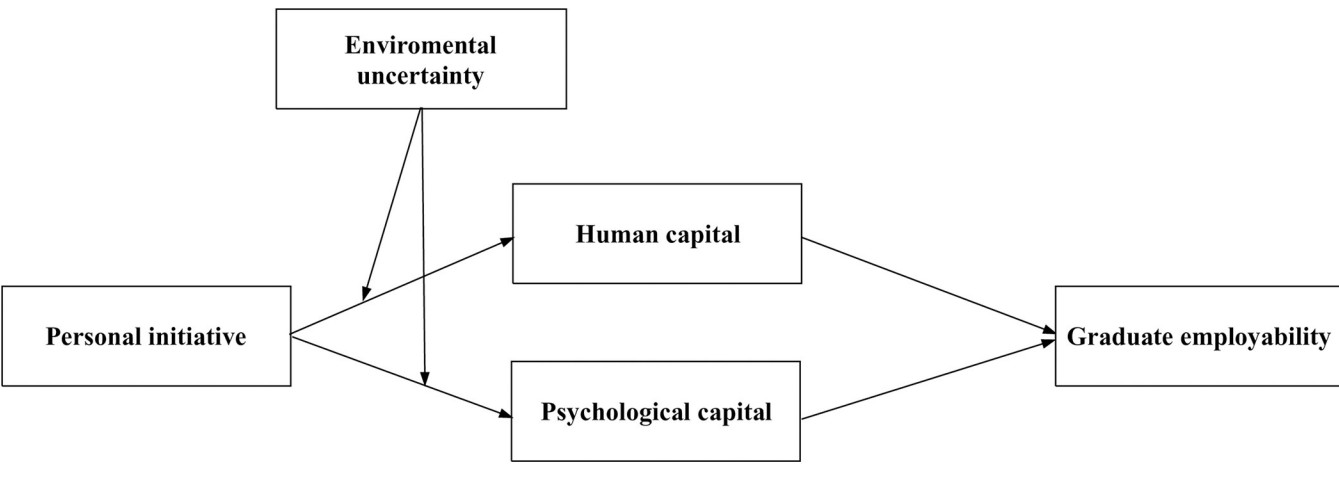

**Fig 1. The dual-path model of personal initiative affecting graduate employability.**

resources (i.e., human and psychological capital) and how environmental uncertainty as particular context moderates the above dual paths. The overarching research model is illustrated in Fig 1.

There are two primary theoretical contributions of this study. The first contribution is the introduction of action theory in graduate employability literature. By considering the development of graduate employability as the action process of goal attachment, the study provides new insight into how PI affects graduate employability through human and psychological capital. The second contribution lies in the investigation of the context of uncertainty. By exploring the moderating effects of environmental uncertainty, the study further clarifies the psychological mechanisms of how graduate employability develops under different conditions and echoes Anderson and Tomlinson's (2021) argument that the context in which university students live could influence their development of graduate employability [28].

## Literature

### PI and graduate employability

Originally developed in the educational psychology, action theory provides at least a partial answer to why some individuals adapt to change and achieve in a better way [29]. The foundational assumption underlying the theory is that action is directed toward the attainment of goals in aspects of sequence, structure, and focus [30]. Specifically, sequence is concerned with how actions unfold in the form of setting goals, processing information of environment, making plans, monitoring the implementation, and responding to feedback. Action theory argues that all goal attainment involves the above-mentioned steps; the more effectively these steps are carried out, the easier it is to achieve the goals. Structure refers to the hierarchical cognitive regulation of action. There are two types of regulations in achieving goals, respectively, task-oriented and metacognitive levels of regulation. Action theory argues that a healthy balance between these two types of regulations makes it possible to show active outcomes. Focus refers to the task and social contexts in which an action is situated and takes place. Action theory argues that individuals need to focus on these contexts to achieve high performance.

According to action theory, university students with PI (characterized by self-starting, proactive, and persistence in overcoming barriers) are more likely to engage in action sequences such as setting higher growth goals, collecting valuable information, and seeking strategies for

difficulties. As one of the most important goals for university students to adapt to the work and professional environment, graduate employability can be broadly defined as the collection of skills, attributes, and characteristics that graduates are expected to be able to demonstrate they have acquired in higher education [31]. New features of organizations where graduates are seeking employment, such as data-driven user responsiveness and cross-boundary resource reconfiguration [e.g., 32,33], have demanded that university students demonstrate a high level of employability at the outset of their careers. As indicated by action theory and research on PI [e.g., 34,35], university students with a high level of PI are more likely to actively search for useful organizational information (e.g., administrative regulations and job qualifications) as they transition from the role of a student to that of an employee. Additionally, they are able to anticipate possible difficulties in their future work and develop solutions for them in advance [36]. More importantly, university students with high PI tend to use multiple strategies to solve problems, including seeking support from senior employees or their superiors [23,37,38]. As a result, more interactions occur in which university students with PI have more opportunities to learn and be recognized than university students without PI. In brief, PI facilitates the action sequence of setting goals, processing information of environment, monitoring the implementation, and responding to feedback, where university students can demonstrate the knowledge, skills, and attributes they have acquired in higher education to their employers. Thus, we propose the following:

**Hypothesis 1:** The PI of university students has a significant positive relationship with their graduate employability.

## Mediating role of capital resource

The PI of university students may have a direct and positive influence on the development of their human capital (HumCap). According to Becker (1993), the HumCap of university students mainly includes the professional knowledge, relevant skills, and social experience they have accumulated through their diverse educational approaches [39]. Compared to high school education, an essential feature of university learning is the ambiguity of the learning goals [40,41]. Many studies have shown that such obstacles can act as goal blockers [e.g., 42,43], preventing university students from excellent performance and ultimately leading to a loss of HumCap. However, university students who are self-starting (one dimension of PI) are well adapted to this mode of university study. According to action theory, these students typically have a high level of metacognitive regulation, by which they can clarify ambiguous learning goals and construct the meaning they find in the process of learning. Meanwhile, proactive university students (another dimension of PI) are more focused on the future. They consciously investigate the competencies required for their intended jobs and undertake targeted learning. These actions taken by university students with high levels of PI contribute to their HumCap development.

Similarly, university students' PI plays an influential role in their development of psychological capital (PsyCap). According to Luthans et al. (2006), PsyCap refers to an individual's positive psychological state of development that is characterized by confidence, optimism, hope, and resiliency. They also proposed a goal-oriented framework that includes goal design, pathway generation, and overcoming obstacles to explain how PsyCap develops [44]. Based on this framework, we can understand how the PI of university students affects their PsyCap effectively. Specifically, PI indicates that the behavior is regulated by goals developed without external pressure, role expectations, or instruction [18]. Thus, PI is the pursuit of self-defined goals rather than specified goals. In this case, university students with PI show more

confidence in the goal design. PI also implies a long-term focus [18]; university students with PI not only focus on potential problems that may arise in the future but also on opportunities. According to action theory, more optimism and hope arise when future opportunities are transformed into goals. In addition, PI usually involves making changes to existing processes, procedures, or tasks that are often frustrating and challenging [18]. For university students with PI, various actionable strategies can be used to cope with those problems as they emerge. In the process, university students develop resilience. In short, PI helps university students develop confidence, optimism, hope, and resilience by methods of goal design, pathway generation, and overcoming obstacles [27,35,45]. Therefore, we propose the following:

**Hypothesis 2a: The** PI of university students has a positive relationship with their HumCap.

**Hypothesis 2b: The** PI of university students has a positive relationship with their PsyCap.

Furthermore, the positive influence of the HumCap/PsyCap of university students on their employability has been widely studied and documented. For example, Clarke (2017) developed a framework to explore the determinants of graduate employability that recognizes the importance of HumCap (e.g., skills, competencies, and work experience) for graduate employability [46]. As another example, Campion et al. (2017) found that the developmental activities specific to students' specific HumCap, such as selecting relevant majors related to their future occupation, make them more qualified for job hunting [47]. Similar results were found in previous research concerning the relationship between PsyCap and employability. Ayala and Manzano (2020) conducted a time-lagged research study, and the results showed that final-year university students' PsyCap has a considerable influence on their employability [48]. Ngoma and Dithan Ntale (2016) further pointed out that the development of university students' PsyCap is beneficial to their graduate employability by improving their social capital, such as motivating their presence on social networks [14].

By synthesizing the above discussion, we can conclude that PI influences graduate employability via two kinds of resource capital, HumCap, and PsyCap. Following the central idea of action theory, university students with PI are more capable of overcoming the difficulties of ambiguous learning objectives in university education and are therefore more likely to accumulate HumCap such as knowledge, skills, and work experience. In return, HumCap, which includes skills that are valuable for future careers, enhances the employability of university students. Simultaneously, university students with PI are more self-driven in setting goals, more opportunity-focused in achieving them, and more action-oriented in facing obstacles, all of which contribute to the development of their psychological capital. PsyCap will ultimately increase the employability of university students. Therefore, we propose the following:

**Hypothesis 3a:** HumCap mediates the relationship between PI and graduate employability.

**Hypothesis 3b:** PsyCap mediates the relationship between PI and graduate employability.

## Moderating role of perceived environmental uncertainty

Action theory also states that the task and social contexts in which an action is situated impose incentives or restrictions on the attainment of goals. University students are at a critical stage of the socialization process [49,50] and gradually learn to recognize the social and employment environment [51,52]. Therefore, How university students perceive the environment outside their universities can critically influence their information-seeking and decision-making behaviors on campus. The extant literature has suggested that the efforts of university students to develop capital resources to enhance their graduate employability are driven not only by personal attributes (e.g., PI) but also by contextual factors [28,53].

An essential feature of university learning is the ambiguity of learning goals, which can prevent university students from gaining HumCap through purposeful learning. This ambiguity is further magnified when university students find themselves in a changing and complicated environment [54–56]. In such an environment, they are pressured to use their limited knowledge or information to build a link between university learning and future employment [57]. Previous research has also shown that most people share a desire for predictability, a preference for order, and discomfort with ambiguity [e.g., 58]; As indicated by action theory, individuals generally tend to adopt actions to reduce uncertainty [29]. As a result, to reduce the unpredictability and chaos that is imposed by environmental uncertainty, university students with high perceived environmental uncertainty are more likely to show behaviors associated with HumCap development, such as more active participation in major courses, more involvement in community activities, and more early engagement in-company internships. In contrast, employees with a low perception of environmental uncertainty are less sensitive to changes or complexity in the external environment, resulting in a lower need to control the environment. In this case, the benefits of PI for graduate employability via HumCap are diminished.

Similarly, the process by which PI influences graduate employability through PsyCap is more likely to occur when the external environment is more uncertain. According to Bae et al. (2022), whether university students take action to improve their future employability depends on whether they judge these actions as necessary. These actions are more likely to occur when the level of uncertainty they experience on campus is higher [59]. However, environmental uncertainty can hurt individuals' PsyCap in the short term by limiting their hope, confidence, and resilience [e.g., 60]. In the long term, however, environmental uncertainty provides opportunities for developing PsyCap [e.g., 61]. To reduce the discomfort caused by uncertainty and to establish a definite connection between the present and the future, university students with a high level of perceived environmental uncertainty tend to set clear, reasonable, and challenging work goals that fully motivate them to learn and work. Moreover, once the goals are defined, these students are more prepared to anticipate possible future obstacles and are more likely to create alternative pathways to address them accordingly. This argument is consistent with the idea of action theory that the most useful way to manage uncertainty is to prepare for it. When taken in an environment of uncertainty, all these actions will contribute to the development of university students' PsyCap, leading to an enhanced indirect effect of PI affecting graduate employability via PsyCap. Therefore, we propose the following:

**Hypothesis 4a:** Perceived environmental uncertainty moderates the indirect effect of PI on graduate employability via HumCap. Specifically, this indirect effect is stronger for university students with high perceptions of environmental uncertainty and weaker for those with low perceptions of environmental uncertainty.

**Hypothesis 4b:** Perceived environmental uncertainty moderates the indirect effect of PI on graduate employability via PsyCap. Specifically, this indirect effect is stronger for university students with high perceptions of environmental uncertainty and weaker for those with low perceptions of environmental uncertainty.

## Method

### Participants and procedures

The study was approved by the Academic Committee of Zhejiang University City College. After the outbreak of the COVID-19 pandemic, we conducted two rounds of questionnaires for university students planning to start working. The first round of questionnaires was

conducted approximately six months before students graduated (Time 1), and the second round was conducted six months after students graduated (Time 2). At Time 1, our participants provided measures for PI, environmental uncertainty, and control variables. At Time 2, the graduated students were asked to rate their HumCap, PsyCap, and graduate employability. received 384 questionnaires at Time 1 and 247 questionnaires at Time 2. The final matched and valid sample consisted of 229 university students.

All participants provided written informed consent related to data use prior to filling out the questionnaires. They were asked to take five minutes to read the introduction of the study, in which we promised that all data would be strictly limited to scientific research and that all personal information would not be leaked to others. To further enhance the anonymity of the data, we asked the university students to fill in the last six digits of their phone numbers and name abbreviations in Pinyin. By doing so, we were able to match the two rounds of questionnaires based on the phone numbers.

## Measures

All the above measures were administered in Chinese and translated from English into Chinese using a standard translation procedure and back-translation procedure to ensure validity [62]. Items regarding the environmental uncertainty were measured on a seven-point rating scale ranging from "1" (strongly disagree) to "7" (strongly agree), while the remaining variables (i.e., PI, HumCap, PsyCap, and graduate employability) were measured on five-point scales ranging from "1" (strongly disagree) to "5" (strongly agree).

**PI.**    We adapted Frese et al.'s (1997) seven-item scale to measure university students' PI [63]. Sample items include the following: "I actively attack problems," "Whenever something goes wrong, I search for a solution immediately," and "Whenever there is a chance to get actively involved, I take it." The coefficient alphas of the scale is .79.

**HumCap.**    In line with Cote and Levine (1997), we used 13 questionnaire items to measure the human capital acquisition of graduates while in school [64]. Participants were asked to rate on questions such as "To what extent do you feel you are acquiring problem-solving skill in your university education". The coefficient alphas of the scale is .83.

**PsyCap.**    To measure it, we adopted a 16-item, shortened version of the psychological capital scale (PCQ-24) [65]. The PCQ-24 scale measures four components of PsyCap, including efficacy, optimism, resilience, and hope. Sample items include: "I feel confident helping to set goals in my learning area during my university education," "I always look on the bright side of things regarding my learning during my university education", "When I have a setback at learning, I easily recover from it during my university education", and "If I should find myself in a jam at learning, I could think of many ways to get out of it during my university education". The coefficient alphas of the scale is .90.

**Environmental uncertainty.**    We adapted Daft et al.'s scale (1988) in accordance with the context of the COVID-19 pandemic to capture the specific environmental uncertainty perceived by university students [66]. Specifically, participants were asked to evaluate the five elements of the environment they faced (three task environments, including the job market, university, and professional skills, and two general environments, including economics and politics) in terms of rate of change and complexity. Sample items include: "Since the outbreak of COVID-19 pandemic, the professional skills required by the market have been changing all the time" (rate of change), and "Since the outbreak of COVID-19 pandemic, the demand for professional skills in the market is high" (complexity). The coefficient alphas of the scale is .94.

**Graduate employability.**    We used an 10-item self-perceived employability scale that was developed by Rothwell and Arnold (2007) to measure graduate employability [67]. Sample

items include: "Even if there was downsizing in this organization, I am confident that I would be retained", "My personal networks in this organization help me in my career", and "The skills I have gained in my present job are transferable to other occupations outside this organization". The coefficient alphas of the scale is .83.

## Method of data analysis

Following previous empirical studies [e.g., 68], we conducted the statistical analysis of the data, including (1) confirmatory factor analysis, (2) tests of convergent and discriminant validity, (3) common method bias analysis, (4) hierarchical regression analysis, (5) analysis of mediation and moderated mediation. To match that, we used a combination of statistical tools such as AMOS, SPSS, and MPLUS.

First, we performed the confirmatory factor analysis and convergent and discriminant validity tests using AMOS. Compared with SPSS, AMOS has two primary advantages [69]. On the one hand, AMOS can provide a series of model fit indices (e.g., $\chi^2/df$, CFI, and RMSEA) to evaluate whether the hypothesized model matches the actual data, whereas SPSS does not. On the other hand, AMOS can specify the relations between the theoretical constructs and their measurement items in a hypothesized model. With AMOS, researchers have access to factor loadings and measurement errors for hypothesized variables, which are prerequisites for calculating convergent and discriminant validity.

Second, we conducted a common method bias analysis, hierarchical regression analysis, and mediation analysis using SPSS. As one of the most used statistical tools, SPSS is more straightforward and effective in running regressions. Research using regression analysis as an empirical approach has been published in many leading academic journals in recent years. In addition, previous research has primarily used the function of exploratory factor analysis (EFA) in SPSS to carry out Harman's single factor test for common method bias [70]. And more importantly, compared to the traditional Sobel test, we can achieve a more accurate estimation of the mediating effect by using the bootstrap sampling method in SPSS PROCESS [71]. In particular, we used 10,000-sample bootstrapping to generate 95% bias-corrected confidence intervals of the mediation effect. If the confidence interval does not contain zero, we can reasonably infer that the mediation effect is significant.

Third, we used MPLUS to examine the moderated mediation effects of environmental uncertainty in the dual-path process of PI affecting graduate employability. Given the extensibility and comprehensiveness of MPLUS, researchers can handle complex models (e.g., our dual-path model of moderated mediation) with a concise syntax, which AMOS or SPSS PROCESS often cannot handle [72]. Specifically, we used MPLUS to quantify the mediation effects and their differences at low (-1SD) and high (+1SD) levels of environmental uncertainty. If the confidence interval of the difference does not contain zero, we can infer that the moderated mediation effect is significant.

## Results

### Confirmatory factor analysis

To provide evidence of construct distinctness, we performed confirmatory factor analysis (CFA) on the survey items from the hypothesized variables: PI, HumCap, PsyCap, environmental uncertainty, and graduate employability. Because of the measurement errors in latent variables caused by too many items, we used the item parceling method to package the items of the variables separately [73], in which PI, environmental uncertainty, and graduate employability each formed three-item packages, and HumCap and PsyCap each formed four-item packages. The standardized loadings of the 17 observed variables on the corresponding latent

**Table 1. Comparison of measurement models.**

| Models | Factors | $\chi^2$ | df | $\chi^2/df$ | RMSEA | CFI | IFI | TLI | SRMR |
|---|---|---|---|---|---|---|---|---|---|
| 1 | *Five factors*: PI, HumCap, PsyCap, environmental uncertainty, and graduate employability | 265.01 | 109 | 2.43 | 0.079 | 0.920 | 0.921 | 0.900 | 0.055 |
| 2 | *Four factors*: Resource capitals (HumCap and PsyCap) combined into one factor. | 342.79 | 113 | 3.03 | 0.094 | 0.882 | 0.883 | 0.857 | 0.063 |
| 3 | *Three factors*: PI and environmental uncertainty combined into one factor; HumCap and PsyCap combined into one factor. | 776.58 | 116 | 6.69 | 0.158 | 0.659 | 0.663 | 0.601 | 0.129 |
| 4 | *Two factors*: Time 1 ratings (PI and environmental uncertainty) combined into one factor; Time 2 ratings (HumCap, PsyCap and graduate employability) combined into one factor. | 838.216 | 118 | 7.10 | 0.164 | 0.629 | 0.632 | 0.572 | 0.132 |
| 5 | *Single factor* | 970.80 | 119 | 8.16 | 0.177 | 0.561 | 0.565 | 0.498 | 0.135 |

*Note*: N = 229.

variables after item packaging ranged from 0.61 to 0.89, with good structural validity. Then, we compared five alternative models with the baseline model, five-factor model 1. As indicated in Table 1, the hypothesized five-factor structure of model 1 with all survey items loading on their respective factors acceptably fit the data with $\chi2/df$ = 2.43, RMSEA = 0.079, CFI = 0.920, IFI = 0.921, TLI = 0.900, and SRMR = 0.055 [74] which provided significant improvement in fit indices compared to the other alternative models (models 2–5). In addition, all standardized factor loadings are above 0.40 ($p < .05$), suggesting that the five constructs captured distinctiveness as expected.

## Tests of convergent and discriminant validity

We used the indicators of average variance extracted (AVE) and composite reliability (CR) to test the convergent validity among the hypothesized variables, and AVE/$r^2$ (comparing AVE and the squared correlations involving the variables) to test the discriminant validity [75]. Taking the variable of PI as an example, we first calculated its AVE. Methodologically, AVE is the average percentage of variation, and the formula to calculate the value is $\Sigma\lambda^2$/n ($\lambda$ = factor loading of every item, n = number of items in a model). By performing CFA in AMOS software, we obtained the factor loading of 0.598, 0.801, and 0.778 for the three packaged items of PI. Then we applied the formula and got the AVE value of PI as 0.54. Similarly, the formula to calculate the value of CR is $\Sigma\lambda^2/(\Sigma\lambda^2 + \Sigma e)$ ($\lambda$ = factor loading of every item, e = error variance of every item), by which we got the CR value of PI as 0.77. Finally, we calculated the values of AVE/$r^2$ between PI and the remaining hypothesized variables. The AVE/$r^2$ values are 4.41 for PI and HumCap, 3.38 for PI and PsyCap, 66.67 for PI and environmental uncertainty, and 7.41 for PI and graduate employability [briefly, (AVE/$r^2 > 1$)].

We tested the convergent and discriminant validity of HumCap, PsyCap, environmental uncertainty, and graduate employability using repeated steps. As shown in Table 2, all the AVE values are above 0.5 (range from 0.52 to 0.75) and CR values are above 0.70 (range from 0.76 to 0.90), indicating good convergent validity for all variables. Moreover, all the AVE/$r^2$ values are above 1 for each variable, indicating good discriminant validity among hypothesized variables.

## Common method bias analysis

Following Podsakoff et al.'s suggestion, we took several measures to reduce the negative influence of common method bias (CMB) [76]. First, we used a time-lagged design to collect data, as the main cause of CMB is getting the data of different variables from the same rater in a short time. Second, prior to filling out the questionnaires, we explained to the participants the purpose of our study, the confidentiality of the information, and the open-ended nature of the

**Table 2. Tests for convergent and discriminant validity.**

| Variables | Convergent validity | Discriminant validity |
|---|---|---|
| PI | | AVE/$r^2$ = [3.38, 66.67] |
| AVE | 0.54 | |
| CR | 0.77 | |
| HumCap | | AVE/$r^2$ = [1.38, 36.80] |
| AVE | 0.53 | |
| CR | 0.82 | |
| PsyCap | | AVE/$r^2$ = [1.40, 95.31] |
| AVE | 0.61 | |
| CR | 0.86 | |
| Environmental uncertainty | | AVE/$r^2$ = [52.08, 300.00] |
| AVE | 0.75 | |
| CR | 0.90 | |
| Graduate employability | | AVE/$r^2$ = [1.35, 208.00] |
| AVE | 0.52 | |
| CR | 0.76 | |

*Note*: N = 229.

responses, in order to encourage them to provide accurate responses as much as possible. Third, we also performed the commonly used Harman's single factor test to check the impact of the CMB on our results. The result shows that the variance for the first factor is 20.78% ($< 40\%$), indicating that the influence of the CMB on the statistical results is not significant.

## Descriptive statistics

Table 3 presents the descriptive statistics and correlations for the proposed variables. As expected, PI was found to be positively related to the graduate employability, HumCap, and PsyCap. Specifically, the Pearson correlation coefficients between PI and graduate employability, HumCap, and PsyCap are 0.27, 0.35, and 0.40, and all significant at the .01 level ($p < .01$). In this case, when the value of PI increases, the values of graduate employability, HumCap, and PsyCap tend to increase. Therefore, our proposed Hypothesis 1, Hypothesis 2a, and Hypothesis 2b are initially verified.

## Hypothesis testing

Table 4 presents the path coefficients and the mediating effects of our research model. As predicted, controlling for the demographic variables, we found that PI is positively related to

**Table 3. Descriptive statistics and inter-correlations among the hypothesized variables.**

| Variables | M | SD | 1 | 2 | 3 | 4 |
|---|---|---|---|---|---|---|
| 1. PI | 3.33 | .66 | | | | |
| 2. HumCap | 3.42 | .52 | .35** | | | |
| 3. PsyCap | 3.40 | .55 | .40** | .66** | | |
| 4. Environmental uncertainty | 3.90 | .73 | .09 | −.12 | −.08 | |
| 5. Graduate employability | 3.12 | .66 | .27** | .62** | .50** | −.05 |

*Note*: N = 229.

* $p < .05$

** $p < .01$.

**Table 4. The direct and mediation effects among PI, HumCap, PsyCap, and graduate employability.**

| The relationships | Direct effect | 95% CI of direct effect, 10000 bootstrap sampling |
|---|---|---|
| PI → Graduate employability (H1) | .25** | CI = [.074 .430] |
| PI → HumCap (H2a) | .26** | CI = [.078 .446] |
| PI → PsyCap (H2b) | .32** | CI = [.095 .537] |
| **The dual paths** | **Mediation effect** | **95% CI of indirect effect, 10000 bootstrap sampling** |
| PI → HumCap → Graduate employability (H3a) | .19** | CI = [.075 .342] |
| PI → PsyCap → Graduate employability (H3b) | .17** | CI = [.067, .315] |

Note: N = 229.

* $p < .05$

** $p < .01$.

graduate employability ($\beta$ = .25**, 95% *CI* = [.074, .430]). Thus, H1 is supported. We also found that PI is positively related to HumCap ($\beta$ = .26**, 95% CI = [.078, .446]) and PsyCap ($\beta$ = .32**, 95% CI = [.095, .537]). Thus, H2a and H2b are supported.

To verify the mediating roles of HumCap and PsyCap, we used bootstrap methods to test their indirect effects through SPSS PROCESS. The results tabulated in Table 4 also showed that the indirect effect of PI on graduate employability via HumCap is .19 (95% CI = [.075, .342]), and the indirect effect of PI on graduate employability via PsyCap is .17 (95% CI = [.067, .315]. With all confidence intervals excluding zero, H3a and H3b are supported.

In H4a and 4b, we proposed that a high-level environmental uncertainty enhances the indirect effects of HumCap and PsyCap between PI and graduate employability, respectively. To understand this in depth, we used a bootstrapping procedure of MPLUS 7.0 [72] to quantify the indirect effects at low (-1SD) and high (+1SD) levels of environmental uncertainty. As shown in Table 5, the indirect effect in Path 1 (PI → HumCap → Graduate employability) is stronger at high ($\gamma$ = .431**, 95% CI = [.271, .611]) rather than low ($\gamma$ = .055, 95% CI = [−.103, .240]) levels of environmental uncertainty, and a further test showed that the difference between these two indirect effects is statistically significant ($\Delta\gamma$ = .376**, CI = [.129, .640]). Similarly, we can note that, in line with H4b, the indirect effect in Path 2 (PI → PsyCap → Graduate employability) is stronger at high ($\gamma$ = .348**, 95% CI = [.220, .499]) rather than low

**Table 5. Conditional indirect effects of PI on graduate employability at values of environmental uncertainty.**

| Moderator | Moderated mediation effect | 95% CI of moderated mediation effect |
|---|---|---|
| Path 1: PI → HumCap → Graduate employability | | |
| Low environmental uncertainty (−1 SD) | .055 | [−.103, .240] |
| High environmental uncertainty (+1 SD) | .431** | [.271, .611] |
| Differ (H4a) | .376** | [.129,.640] |
| Path 2: PI → PsyCap → Graduate employability | | |
| Low environmental uncertainty (−1 SD) | .074 | [−.074, .258] |
| High environmental uncertainty (+1 SD) | .348** | [.220, .499] |
| Differ (H4b) | .274* | [.076, .490] |

*Note.* N = 229.

* $p < .05$

** $p < .01$.

($\gamma$ = .074, 95% CI = [−.074, .258]) levels of environmental uncertainty, and a further test showed that the difference between these two indirect effects is statistically significant ($\Delta\gamma$ = .274*, CI = [.076, .490]). To summarize, these results support H4a and H4b.

## Discussion

In the present study, we proposed a dual-path model to investigate how PI promotes graduate employability in the context of environmental uncertainty. The study's main findings are as follows. First, the PI of university students has a positive effect on their graduate employability. Second, the PI of university students has a positive effect on their HumCap and PsyCap. Third, both HumCap and PsyCap play mediating roles in the relationship between PI and graduate employability. Fourth, environmental uncertainty plays a moderating role in the above mediating processes. Specifically, a high level of environmental uncertainty strengthens the positive indirect effects by which PI affects graduate employability through HumCap (PsyCap). These findings contribute to the existing literature on graduate employability and have crucial implications for higher education institutions and government policy-makers.

### Theoretical and practical contributions

The main theoretical contributions of our study are twofold. First, the study provides new insight into the development of graduate employability by introducing action theory. For university students, the development of graduated employability is considered as an important goal, and therefore the attachment of the goal is a series of action processes. From this aspect, we introduced the PI, a promising construct in action theory, into the research model. At the individual level, although scholars have studied the factors of academic achievements [12], soft skills [13], and career identity [14] on graduate employability, few of them have emphasized the impact of PI. Our findings on the relationship between PI and graduate employability extend the contextualized research of PI in higher education research and add to the empirical studies on graduate employability, thus contributing to the cross-fertilization of different perspectives in the research domains of PI and graduate employability.

Moreover, our study explores the dual process of PI affecting graduate employability based on HumCap and PsyCap and therefore further reveals the psychological mechanisms of developing graduate employability. For university students, HumCap and PsyCap are two important sources of their sustainable competitive advantage [e.g., 31,77], yet most previous studies tended to explore their impacts on graduate employability in isolation [48,64]. To our best of knowledge, our study is one of the few studies that effectively integrates two crucial capital resources in the graduate employability literature. With action theory, our study combines PI, capital resources, and graduate employability in an integrated way, thus providing a more comprehensive insight into the development of graduate employability.

Second, by introducing the moderator of environmental uncertainty, the present study also makes an important contribution to the literature in understanding how graduate employability develops in a dynamic context. To our knowledge, this study is one of the few that examines the moderating effects of environmental uncertainty in fostering graduate employability. As noted, most previous studies in this domain were conducted in stable settings [77], and thus, their findings cannot be extended to uncertain contexts (e.g., the COVID-19 pandemic) without empirical testing. By considering the role of environmental uncertainty on the two alternative pathways, we found that environmental uncertainty can serve as a favorable context that strengthens the positive indirect effect(s) of PI affecting graduate employability. This finding further clarifies the psychological mechanisms of how graduate employability develops under different conditions and echoes Anderson and Tomlinson's (2021) argument that the context

in which university students live could influence their development of graduate employability [28].

Our study also has some practical implications. One of the critical challenges that higher education institutions or government policy-makers face is to address the mismatch between low graduate employability and the growing demands of the labor market [9], especially during the COVID-19 pandemic, when most companies have scaled back their hiring. Our findings suggest that such challenges could be overcome by teaching university students PI before they search for jobs. By providing a psychology-based PI training program [27], higher education institutions or government policy-makers can equip university students with a self-starting, future-oriented, and persistently proactive mindset to help them develop the capital resources employers expect and stand out in the job market. Meanwhile, our findings also suggest that uncertainty is not a bad thing; a perception of high environmental uncertainty is helpful for university students' development of graduate employability. Therefore, universities must not become the "ivory tower" that isolates students from society but rather inform them of the complexity and rapid changes in the current external environment through various means, such as classroom lectures or academic seminars. By doing so, university students will be inspired to act more proactively to reduce the stress caused by uncertainty, and in turn, their graduate employability is more likely to be developed.

## Limitations and future research

Several potential limitations of the present study should be noted. First, the social desirability problem in measuring PI may have biased our results. Because of the tendency to maintain self-esteem or to create a good impression, individuals generally exaggerate their positive traits (e.g., PI) when filling out the questionnaires. Although we used a time-lagged research design to improve the reliability of our study, the self-reported data we obtained may still overestimate the actual PI of the participants [e.g., 78], leading to biased results. Therefore, future research using peer assessment or more objective methods is needed to measure PI and further validate our findings. For example, researchers may invite university students' mentors, roommates, or classmates who are familiar with them to rate their PI, or they could design a quasi-experimental study that teaches and promotes PI, by which researchers would be able to observe university students' PI more accurately.

Second, although we advanced the graduate employability literature by building a two-path model that introduces intrapersonal resources (i.e., HumCap and PsyCap) to explain the formation process of graduate employability, we may ignore the important mediating role of social capital in understanding how PI affects graduate employability [e.g., 79]. As an interpersonal resource, social capital has been shown to be related to PI [e.g., 80] and graduate employability [e.g., 31,79] separately in previous research. However, the exclusion of social capital in our dual-path model may prevent us from developing a more accurate and comprehensive picture of how graduate employability is fostered. Therefore, future research can draw on social exchange theory to validate whether PI affects graduate employability via social capital.

## Conclusion

Overall, the present study advanced our understanding of how university students' personal initiative affects their graduate employability and how the process unfolds in a specific context of uncertainty by introducing action theory. Using a time-lagged design, we found that personal initiative positively influences graduate employability by promoting university students' human and social capital. We also introduced the COVID-19 pandemic as a context of uncertainty and found that a high-level perceived environmental uncertainty plays a favorable role

in developing graduate employability. As such, government and universities should consider providing PI training programs and creating an environment of uncertainty for university students as a starting point for improving their employability.

## Supporting information

**S1 Data.**
(ZIP)

## Author Contributions

**Conceptualization:** Ting Wu.

**Formal analysis:** Ting Wu.

**Funding acquisition:** Nan Ma.

**Investigation:** Ting Wu.

**Supervision:** Ting Wu, Nan Ma.

**Validation:** Qin Lai.

**Writing – original draft:** Ting Wu, Qin Lai, Nan Ma.

**Writing – review & editing:** Ting Wu, Nan Ma, Yixuan Shao.

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
