## [Decision Letter · Decision Letter 0]

1 Jun 2022

PONE-D-22-11292Surviving Uncertainty: A Dual-Path Model of Personal Initiative Affecting Graduate EmployabilityPLOS ONE

Dear Dr. Ma,

Thank you for submitting your manuscript to PLOS ONE. After careful consideration, we feel that it has merit but does not fully meet PLOS ONE’s publication criteria as it currently stands. Therefore, we invite you to submit a revised version of the manuscript that addresses the points raised during the review process.

We look forward to receiving your revised manuscript.

Kind regards,

Sathishkumar V E

Academic Editor

PLOS ONE

Journal Requirements:

- https://journals.plos.org/plosone/article?id=10.1371%2Fjournal.pone.0251850

In your revision ensure you cite all your sources (including your own works), and quote or rephrase any duplicated text outside the methods section. Further consideration is dependent on these concerns being addressed.

Reviewers' comments:

Reviewer's Responses to Questions

**Comments to the Author**

1. Is the manuscript technically sound, and do the data support the conclusions?

Reviewer #1: No

Reviewer #2: Yes

2. Has the statistical analysis been performed appropriately and rigorously? 

Reviewer #1: No

Reviewer #2: Yes

3. Have the authors made all data underlying the findings in their manuscript fully available?

Reviewer #1: No

Reviewer #2: Yes

4. Is the manuscript presented in an intelligible fashion and written in standard English?

Reviewer #1: No

Reviewer #2: Yes

5. Review Comments to the Author

Reviewer #1: 1.Introduction section needs to be re-written to improve its quality and readability.

2.What is the motivation of the proposed work? Research gaps, objectives of the proposed work should be clearly justified

3.Overall, the basic background is not introduced well, where the notations are not illustrated much clear.

4.The literature has to be strongly updated with some relevant and recent papers focused on the fields dealt with in the manuscript.

5.The study lacks a theoretical framework which is important for the reader to grasp the crust of the research.

6. Explain why the current method was selected for the study, its importance and compare with traditional methods.

7.Authors are suggested to include more discussion on the results and also include some explanation regarding the justification to support why the proposed method is better in comparison towards other methods

8.Does this kind of study have never attempted before? Justify this statement and give an appropriate explanation to do so in this paper.

9.The language usage throughout this paper need to be improved, the author should do some proofreading on it.

Reviewer #2: English langiuage check needed.

Add conclusion section summarizing the result of the study

Add the contributions of research as points at the end of introduction section

Explain the correlations values in Table 3.

Table 2: Explain in Detail

6. PLOS authors have the option to publish the peer review history of their article (what does this mean?). If published, this will include your full peer review and any attached files.

Reviewer #1: No

Reviewer #2: **Yes: **Usha Moorthy

---

## [Author Response · Author response to Decision Letter 0]

16 Jun 2022

Please check our responses in the "Response to Reviewers" file. Thank you.

---

## [Decision Letter · Decision Letter 1]

20 Jun 2022

Surviving Uncertainty: A Dual-Path Model of Personal Initiative Affecting Graduate Employability

PONE-D-22-11292R1

Dear Dr. Ma,

We’re pleased to inform you that your manuscript has been judged scientifically suitable for publication and will be formally accepted for publication once it meets all outstanding technical requirements.

Kind regards,

Sathishkumar V E

Academic Editor

PLOS ONE

Additional Editor Comments (optional):

Reviewers' comments:

Reviewer's Responses to Questions

**Comments to the Author**

1. If the authors have adequately addressed your comments raised in a previous round of review and you feel that this manuscript is now acceptable for publication, you may indicate that here to bypass the “Comments to the Author” section, enter your conflict of interest statement in the “Confidential to Editor” section, and submit your "Accept" recommendation.

Reviewer #1: All comments have been addressed

Reviewer #2: (No Response)

2. Is the manuscript technically sound, and do the data support the conclusions?

Reviewer #1: Partly

Reviewer #2: (No Response)

3. Has the statistical analysis been performed appropriately and rigorously? 

Reviewer #1: No

Reviewer #2: (No Response)

4. Have the authors made all data underlying the findings in their manuscript fully available?

Reviewer #1: (No Response)

Reviewer #2: (No Response)

5. Is the manuscript presented in an intelligible fashion and written in standard English?

Reviewer #1: Yes

Reviewer #2: (No Response)

6. Review Comments to the Author

Reviewer #1: After carefully checking, I confirm that the authors have completely addressed all my concerns, I therefore recommend accepting this paper for publication

Reviewer #2: (No Response)

7. PLOS authors have the option to publish the peer review history of their article (what does this mean?). If published, this will include your full peer review and any attached files.

Reviewer #1: No

Reviewer #2: **Yes: **Usha Moorthy

---

## [Editor Report · Acceptance letter]

30 Jun 2022

PONE-D-22-11292R1 

Surviving Uncertainty: A Dual-Path Model of Personal Initiative Affecting Graduate Employability 

Dear Dr. Ma:

I'm pleased to inform you that your manuscript has been deemed suitable for publication in PLOS ONE. Congratulations! Your manuscript is now with our production department. 

Kind regards, 

on behalf of

Dr. Sathishkumar V E 

Academic Editor

PLOS ONE